# Learning High-Order Motion Patterns from Event Stream for Continuous Space-Time Video Super-Resolution

## Abstract

Current methods in the domain of continuous space-time video super-resolution achieve temporal alignment by predicting motion between frames. However, these frame-based approaches encounter challenges with inaccurate optical flow estimation. To overcome this, we incorporate event data, enhancing both temporal and spatial aspects of video super-resolution. Based on the motion details conveyed by event streams, our proposed method, EvTaylor-Net, performs a Taylor expansion approximation of the object motion function at specified timestamps to estimate more precise forward optical flow. Our method estimates the masks from the event surface to alleviate the issue of multiple source pixels mapping to the same target position during the forward warping process. Furthermore, EvTaylor-Net adopts local implicit neural representation to simultaneously enhance the resolution of videos in both temporal and spatial domain, ensuring a comprehensive improvement of video quality. Extensive experimental results demonstrate that the proposed EvTaylor-Net, bolstered by event streams, outperforms state-of-the-art methods for spatio-temporal video super-resolution tasks.

## 1 Introduction

Continuous Space-Time Video Super-Resolution (C-STVSR) task aims to simultaneously and arbitrarily increase spatial and temporal resolutions of low-resolution (LR) and low-frame-rate (LFR) video sequences. In comparison to Fixed-scale Space-Time Video Super-Resolution (F-STVSR) task Geng et al. (2022); Haris et al. (2020); Xiang et al. (2020); Xu et al. (2021b); Huang et al. (2023) which preforms space-time video super-resolution at predetermined spatio-temporal scales, C-STVSR approaches are more flexible and practical in real-world scenarios, serving as a comprehensive paradigm.

Great success has been recently achieved in C-STVSR task. A pioneering approach VideoINR Chen et al. (2022b) sequentially learn spatial and temporal local implicit neural representation (INR) for super-resolving and interpolating the frame features, achieving a continuous space-time video super-resolution. However, learning backward motion through temporal implicit neural functions is challenging. For example, in Fig. 1 (b), the backward motion displacement of point $p_3$ at $t = 1$ and $t = 2$ are determined by two separate motion trajectories originating from points $p_2$ and $p_3$ in the reference frame at $t = 0$, respectively. Essentially, when the backward motion vectors at $p_3$ are observed over time, they represent a combination of various motion trajectories. This complexity might lead to unwanted randomness and inconsistencies in the temporal INR (T-INR). The limitations associated with backward motion estimation have been thoroughly examined in MoTIF Chen et al. (2023). The authors propose a novel approach that prioritizes the forward motion of pixels (illustrated in Fig. 1 (c)). This method inherently captures continuous motion trajectories, which significantly enhances its suitability for INR learning. However, since the initial optical flow is derived from a pre-trained estimator, there are limitations that cause the optical flow estimation to be potentially inaccurate, e.g., in Fig. 1 (c), the initial optical flow (blue arrow) at $t = 2$ is inaccurate , which causes the optical flow estimation at any time to become worse (red arrow). To overcome this, as illustrated in Fig. 1 (d), our research incorporates event streams for more precise optical flow estimation, aiming to enhance the overall accuracy and efficacy of C-STVSR task.

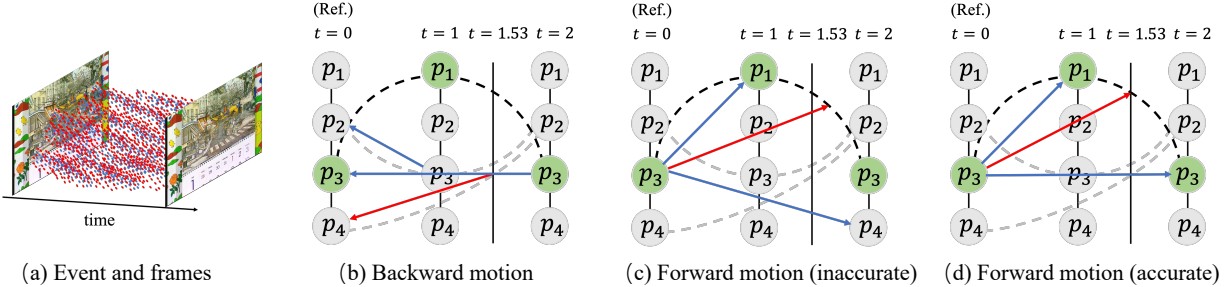

Figure 1: Illustration of different motion estimation in C-STVSR methods. The circles represent the pixel positions in the given video. Dashed lines indicate the motion trajectories of the pixels. Blue arrows are the displacement of backward/forward motion, i.e optical flow. Red arrows show the optical flow at an arbitrary timestamp estimated based on the blue arrows. (b) demonstrates the traditional backward motion estimation method Chen et al. (2022b), (c) shows the forward motion estimation manner in MoTIF Chen et al. (2023). We introduce the concurrent event stream between frames, as shown in (a), and further propose an accurate forward motion estimation, as illustrated in (d).

The event camera, as a novel bio-inspired camera, can asynchronously record pixel-level brightness changes in the format of streams. With the inherent properties of high temporal resolution, high dynamic range, and low latency, event cameras have demonstrated their potential as a complement to traditional frames in various computer vision tasks, including image deblurring Wang et al. (2020); Xu et al. (2021a); Song et al. (2022); Chen et al. (2022a); Kim et al. (2022), high dynamic range (HDR) imaging Han et al. (2020; 2023); Wang et al. (2021); Zou et al. (2021), spatial super-resolution Jing et al. (2021); Wang et al. (2020); Han et al. (2021); Lu et al. (2023), and VFI Sun et al. (2023); Tulyakov et al. (2022; 2021); He et al. (2022).

Given the high temporal resolution of event streams and their benefits for spatial information reconstruction, event cameras have significant potential to simultaneously enhance the temporal and spatial resolution of videos. Therefore, this paper introduces an INR-based network, named EvTaylor-Net, that harness the advantages of event streams for the C-STVSR task. The proposed EvTaylor-Net takes two consecutive LR images and the concurrent event streams as input, and outputs intermediate high-resolution (HR) frames at arbitrary spatio-temporal upscaling factors. To better align temporal features, this paper extracts motion details between frames from event streams for high-order motion modeling. Diverging from previous research that relied on linear Bao et al. (2019b); Yuan et al. (2019); Niklaus & Liu (2018); Chen et al. (2023) quadratic Xu et al. (2019); Liu et al. (2020) and cubic Chi et al. (2020) motion assumptions to estimate the optical flow, our model employs Taylor expansion approximations at specific timestamp on the object motion function. This approach extends the traditional motion assumptions to $nth$ ($n \geq 1$) order motion, enabling more precise forward optical flow estimation. Forward warping methods inevitably encounter cases where multiple pixels map to the same target location. Prior studies have demonstrated that assigning weights to the significance of different pixels (mainly by their depth), can effectively address this issue Wang et al. (2018); Niklaus & Liu (2020a). In this paper, we estimate the important weighted maps (i.e, masks) based on the frequency of event streams. We choose frequency information because different objects, due to their speed, shape, and texture variations, trigger events at varying rates. Event streams help distinguish the location of different objects and thus the foreground from the background.

In summary, the contributions of this paper are as follows:

1. We are the first to integrate the event streams into the C-STVSR task. The proposed **EvTaylor-Net**, harnesses **Ev**ent data and utilizes **Taylor** expansion approximations to estimate the high-order motion between two consecutive LR frames.

2. We design an occlusion estimation network based on varying event frequencies to estimate importance masks, which alleviate the challenge of mapping multiple sources to the same target position.

3. Extensive experiments on public benchmark datasets demonstrate the superiority of the proposed EvTaylor-Net, which overs state-of-the-art methods in both temporal and spatial video super-resolution.

## 2 Related Work

### 2.1 Event-guided Video Frame Interpolation & Video Super-Resolution

VFI Cheng & Chen (2021); Lee et al. (2020); Niklaus et al. (2021); Huang et al. (2022); Kong et al. (2022); Li et al. (2023) is a kind of task that extracts the high frame rate video from the LFR video. Recently, with the emergence of event cameras, event-guided VFI approaches Tulyakov et al. (2021; 2022); Yu et al. (2021); He et al. (2022); Lin et al. (2020); Zhang & Yu (2022); Kim et al. (2023); Sun et al. (2023); Gao et al. (2022) have attracted significant attention, primarily due to the microsecond-level time resolution inherent in event streams. For instance, TimeLens Tulyakov et al. (2021) pioneered the introduction of a VFI model that incorporates both warp-based and composition-based methodologies. CBMNet Kim et al. (2023) proposed an event-based VFI framework with cross-modal asymmetric bidirectional motion field estimation, achieving competitive performance.

VSR approaches Caballero et al. (2017); Tao et al. (2017); Sajjadi et al. (2018); Xue et al. (2019); Chan et al. (2021); Li et al. (2023) aim to increase the spatial resolution for the given low-resolution video. Since event streams with high temporal resolution offer critical insights for spatial super-resolution, event data has been increasingly utilized to enhance video super-resolution processes Jing et al. (2021); Lu et al. (2023). Jing et al. Jing et al. (2021) introduced an E-VSR framework that extracts super-resolved information from the event stream with high temporal resolution, enabling the recovery of high-resolution frames from low-resolution counterparts. As the E-VSR Jing et al. (2021) model is limited to a fixed upscale factor, Lu et al. Lu et al. (2023) further proposed a continuous VSR framework based on INR methods, enabling VSR at arbitrary up-scaling factor. Unlike the previously mentioned approaches, which exclusively use event data to guide VFI or VSR tasks, our work aims to address continuous space-time video super-resolution tasks simultaneously.

### 2.2 Space-Time Video Super-Resolution

Space-Time Video Super-Resolution (STVSR) tasks aim to increase both the spatial and temporal resolution of a given video. In general, STVSR approaches can be categorized into two categories: two-stage STVSR and one-stage STVSR methods. The two-stage STVSR methods involve cascading the VFI and VSR subnetworks to successively improve temporal and spatial resolution. However, these methods face challenges in effectively handling the internal correlation between spatial and temporal features. To address this, recent works Xiang et al. (2020); Xu et al. (2021b); Geng et al. (2022); Haris et al. (2020); Huang et al. (2023) have started employing the one-stage method to enhance spatial and temporal resolution simultaneously. However, these approaches are primarily focused on fixed-scale STVSR. More recently, VideoINR Chen et al. (2022b) exploits the INR method to allow the continuous spatial and temporal scales for STVSR tasks. Building upon this, MoTIF Chen et al. (2023) introduces a novel approach by advocating for the learning of forward motion as opposed to backward motion. This preference is rooted in the fact that forward motion encapsulates a continuous motion trajectory, which inherently aligns more effectively with the principles of INR learning. However, the optical flow predicted by MoTIF with a pre-trained optical flow estimator is not suitable for all datasets. Addressing this, we import the event data to assist the optical flow estimation, and generalize motion to the $nth$ order paradigm via Taylor expansion approximation. To the best of our knowledge, DTEA Zhu et al. (2023) and PDTE Zhu et al. (2024) employ Taylor expansion approximations to estimate optical flow between two consecutive frames. However, their approach, which uses the pixel $x$ as the variable in the Taylor expansion, presents challenges in the physical interpretation of higher-order derivatives. Furthermore, these methods lack the incorporation of supplementary data to enhance the accuracy of Taylor expansion approximations.

## 3 Methodology

### 3.1 Formulation

**Event camera model.** The event camera detects the logarithmic brightness change asynchronously in each pixel. When the brightness changes over a threshold $\theta$, an event will be triggered. The event streams $\mathcal{E}$ triggered between LR frames $I_0^L$ and $I_1^L$ are formatted as a set of quartets $(e_x, e_y, t, p)$. Here, $e_x$ and $e_y$

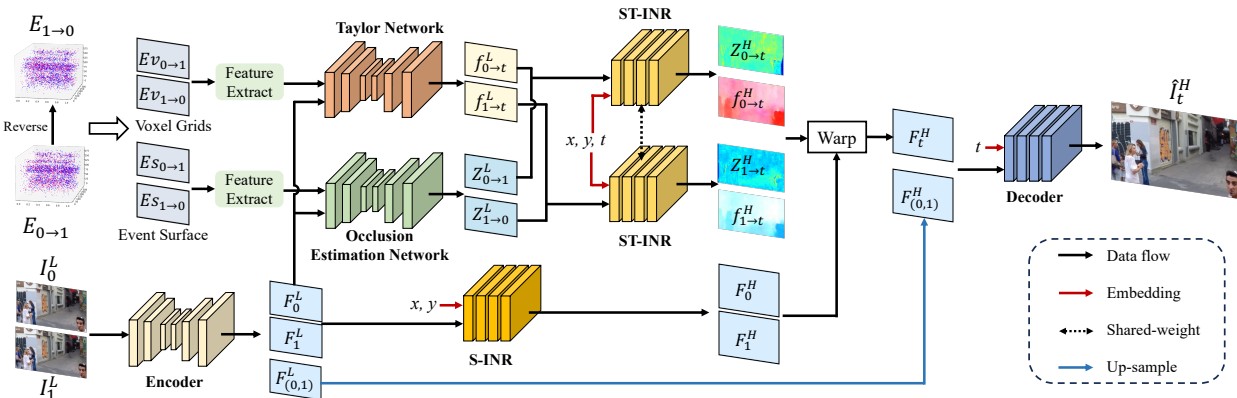

Figure 2: Network architecture of EvTaylor-Net. The encoder network processes the provided LR frames to extract their features. The original event $E_{0\to1}$ and the reversed event $E_{1\to0}$ are represented as event surface and event voxel grid. The Taylor network, utilizing the features from voxel grids along with image features $F_0^L$ and $F_1^L$, estimates the features of the bi-directional optical flow $f_{0\to t}^L$ and $f_{1\to t}^L$. Meanwhile, the occlusion estimation network, fed with the event surface and image features, predicts the mask features $Z_{0\to1}^L$ and $Z_{1\to0}^L$. The S-INR network recovers the image features of the target space size $F_0^H$ and $F_1^H$. The ST-INR network learns HR optical flow and masks. Softmax splatting operation facilitates forward warping to acquire target features $F_H^t$, and the decoder is tasked with reconstructing the RGB HR image from these HR features.

denote the coordinates of the event signal. $t$ represents a timestamp indicating when an individual event occurred, falling within the interval $[0,1]$. $p \in \{+1,-1\}$ signifies the polarity of the event signal. A positive polarity indicates an increase in logarithmic brightness, whereas a negative polarity signifies a decrease.

**Continuous STVSR with event data.** Given two LR frames $I_0^L$ and $I_1^L$, along with a concurrent event stream $\mathcal{E}$, our model aims to recover the intermediate HR frames $\hat{\mathbf{I}}^{\mathbf{H}}$. This process can be represented by the function $\mathcal{F}$, defined as:

$$\hat{\mathbf{I}}^{\mathbf{H}} = \{\hat{I}_t^H | t = t_1, t_2, \cdots, t_n\} = \mathcal{F}(I_0^L, I_1^L, \mathcal{E}), \tag{1}$$

where $t_1$ to $t_n$ represent any group of increasing timestamps in the interval [0,1], with $n \geq 1$.

In terms of the continuous spatial super-resolution, given the spatial coordinate $x_s$, the Spatial local Implicit Neural Representation (S-INR) network newtork, denoted as $\mathcal{F}_s$, is responsible for mapping the features of LR frames into HR spatial features corresponding to $F_0^H$ and $F_1^H$. This process can be represented by the following equation:

$$\{F_0^H, F_1^H\} = \mathcal{F}_s(I_0^L, I_1^L, x_s). \tag{2}$$

As for the temporal super-resolution, we employ the event stream $\mathcal{E}$, comprising both the forward event $E_{0\to1}$ and the reversed event $E_{1\to0}$ to accurately estimate optical flow in LR dimensions. This estimation employs Taylor expansion approximation, denoted as $\mathcal{T}$. Given the space-time coordinate $x_{st}$, the Spatial-Temporal local Implicit Neural Representation (ST-INR) network, $\mathcal{F}_{st}$, is used to estimate the motion field in HR dimensions. The process can be mathematically formulated as:

$$\{f_{0\to t}^H, f_{1\to t}^H\} = \mathcal{F}_{st}(f_{0\to t}^L, f_{1\to t}^L, x_{st}) = \mathcal{F}_{st}(\mathcal{T}(I_0^L, E_{0\to1}, I_1^L, E_{1\to0}), x_{st}). \tag{3}$$

Additionally, the event stream $\mathcal{E}$ plays a crucial role in estimating the masks $Z_{0\to1}^L$ and $Z_{1\to0}^L$ for forward warping operations. This estimation is facilitated by the occlusion estimation function, designated as $\mathcal{O}$. Utilizing the ST-INR network, we are able to obtain these occlusion masks in HR dimensions. This process can be mathematically expressed as:

$$\{Z_{0\to t}^H, Z_{1\to t}^H\} = \mathcal{F}_{st}(Z_{0\to1}^L, Z_{1\to0}^L, x_{st}) = \mathcal{F}_{st}(\mathcal{O}(I_0^L, E_{0\to1}, I_1^L, E_{1\to0}), x_{st}). \tag{4}$$

The target HR frames features can be obtain by warping the HR features of reference frames according to the optical flow. Subsequently, the decoder $\mathcal{D}$ generates the HR frames in the RGB domain: Thus, the

intermediate HR frame at arbitrary time $t$ can be recovered:

$$I_t^H = \mathcal{D}(\mathcal{W}(F_0^H, f_{0 \to t}^H, Z_{0 \to t}^H) + \mathcal{W}(F_1^H, f_{1 \to t}^H, Z_{1 \to t}^H)), \tag{5}$$

where $\mathcal{W}$ indicates the softmax splatting Niklaus & Liu (2020b) operation.

### 3.2 Proposed Framework

As illustrated in Fig. 2, the proposed EvTaylor-Net takes two consecutive LR frames, $I_0^L$ and $I_1^L$, together with the event stream $\mathcal{E}$ captured between them, as inputs to recover intermediate HR frames $\hat{\mathbf{I}}^{\mathbf{H}}$. Both the input LR frames and event data have a spatial resolution of $H \times W$. The output frames are of a spatial size $H' \times W'$. The spatial scaling factor is defined as $s = W'/W = H'/H \geq 1$.

**Event representation.** We employ two distinct representations: the event voxel grids and the event surface. The conventional methods Zhu et al. (2019); Rebecq et al. (2019) for constructing event voxel grids involves mapping the time span between the first and last events into a range of $[0, B - 1]$, where $B$ represents the number of temporal intervals. However, this construction method may introduce ambiguity, as it regards the starting and ending times to be the timestamps of the first and last events, rather than those of the reference images (*e.g.*let@tokeneonedot, $I_0^L$ and $I_1^L$). This discrepancy in starting and ending times may lead to two distinct event streams mapped to the same voxel grids. Recognizing the importance of motion at times 0 and 1 for optical flow estimation in the proposed Taylor network, we partition the time duration $[0, 1]$ between $I_0^L$ and $I_1^L$ into $B$ temporal intervals to avoid this ambiguity. As for events occurring between the reference images $I_0^L$ and $I_1^L$, we transform them into the format of event voxel grids according to their polarity:

$$E(e_x, e_y, t) = \sum_i p_i \max(0, 1 - |t - t_i^*|), \tag{6}$$

where $p_i \in \{-1, +1\}$, and $t_i^* = (B - 1) \times t_i$. In this way, we can obtain the $Ev_{0 \to 1}$ of size $(H, W)$ with $2 \times B$ channels. In the context of event surface representation, we partition the normalized time interval $[0, 1]$ into $B$ intervals. We aggregate the triggered event count in the time interval $[0, i/B]$ ($i \in [1, B]$) and record the timestamp of the last event at the pixel-level. Similar with event representation manner in the works Park et al. (2016); Wu et al. (2022); He et al. (2022), the event surface representation of each bin includes four-channels, the first two channels represent the positive and negative event counts within the temporal interval, while the third and fourth channels store the timestamps of the last events. This process yields the $Es_{0 \to 1}$ matrix of size $(H, W)$ with $4 \times B$ channels. To acquire the bi-directional optical flow for temporal super-resolution, we reverse both the time order and polarity of the original event stream $E_{0 \to 1}$, resulting in a backward event stream $E_{1 \to 0}$. We then apply the same representation method to the reversed event stream to obtain event voxel grids $Ev_{1 \to 0}$ and the event surface $Es_{1 \to 0}$.

**Taylor network for optical flow estimation.** To obtain latent frames at arbitrary timestamp, the proposed EvTaylor-Net leverages the complementary information from both image and event data to predict bi-directional optical flow. The event stream captured between frames is rich in motion cues, significantly aiding in the modeling of high-order motion. Let's assume that the motion of objects is represented by the function $F(t)$, where time serves as the input and the object's position as the output. Employing an infinite-order Taylor's series expansion, we can articulate this relationship as follows:

$$F(t) = F(t_0) + \frac{F'(t_0)}{1!}(t - t_0) + \cdots + \frac{F^{(k)}(t_0)}{k!}(t - t_0)^k + \cdots. \tag{7}$$

When only considering the *nth* order Taylor's approximations, the formula simplifies to:

$$F(t) = F(t_0) + \sum_{k=1}^{n} \frac{F^{(k)}(t_0)}{k!}(t - t_0)^k. \tag{8}$$

Optical flows between two frames are the displacement between them, i.e, the difference in the output of the object motion function $F(t)$ at two distinct moments. Thus, by directly manipulating the aforementioned

equation, we can derive the optical flow from time stamp $t_0$ to $t$:

$$f_{t_0 \to t} = F(t) - F(t_0) = \sum_{k=1}^{n} \frac{F^{(k)}(t_0)}{k!}(t - t_0)^k. \tag{9}$$

This implies that the optical flow $f_{t_0 \to t}$ can be represented as the weighted sum of $1st$ order to $nth$ order of the object motion function $F(t)$ at timestamp $t_0$. In this paper, we aims to obtain the bi-directional optical flow, leading us to define the forward motion function as $F_{fw}(t)$ and the backward motion function as $F_{bw}(t)$. Regarding the forward optical flow, we estimate $t_0$ at time 0 (i.e, the timestamp of $I_0^L$). Following Eq. 9, the forward optical flow $f_{0 \to t}^L$ can be expressed as:

$$f_{0 \to t}^L = \sum_{k=1}^{n} \frac{F_{fw}^{(k)}(0)}{k!} t^k. \tag{10}$$

In terms of the backward optical flow, we estimate $t_0$ at time 1, i.e, the timestamp of $I_1^L$. Therefore, we can calculate the backward optical flow $f_{1 \to t}^L$:

$$f_{1 \to t}^L = \sum_{k=1}^{n} \frac{F_{bw}^{(k)}(1)}{k!} (1 - t)^k. \tag{11}$$

Following Eq. 10 and Eq. 11, we introduce a Taylor network designed to estimate the bi-directional optical flows $f_{0 \to t}^L$ and $f_{1 \to t}^L$. Fig. 3 provides an illustration of the Taylor network's operation in estimating $f_{0 \to t}^L$. This process involves feeding features from the event voxel grid $Ev_{0 \to 1}$ and the image $I_0^L$ into a fusion network, which constructs the 0th-order feature at time 0. The derivative function, implemented by a U-Net (referred to as G here), takes $F_{fw}^{(k-1)}(0)$ as input and produces its derivative $F_{fw}^{(k)}(0)$. The derivative function G is weight-shared, with each output $F_{fw}^{(k)}(0)$ in every iteration being scaled by the coefficient $t^k/k!$. Summing all these outputs results in the optical flow features $f_{0 \to t}^L$. In a similar fashion, the estimation

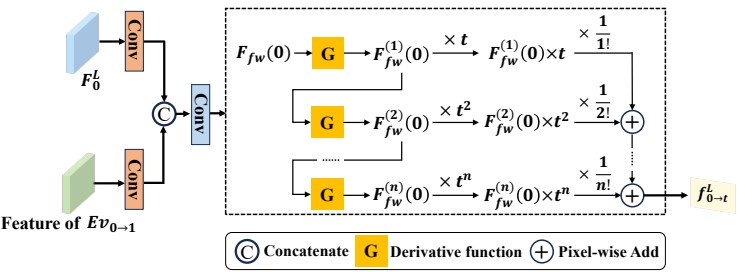

Figure 3: Network architecture of Taylor network to estimate $f_{0 \to t}^L$. It extracts the 0th-order feature of $F_{fw}(0)$ according to the image feature $F_0^L$ and the feature of forward event voxel grids $Ev_{0 \to 1}$. The shared-weight module G is utilized for the derivation operation. The optical flow features are computed from the weighted summation of the output of each order of derivative.

for the backward optical flow $f_{1 \to t}^L$ adheres to the same structure. It takes the feature of $Ev_{1 \to 0}$ and $F_1^L$ as input, while the derivative function G takes $F_{bw}^{(k-1)}(1)$ as input and produces its derivative $F_{bw}^{(k)}(1)$. The network details of the derivative function G is provided in the Appendix.

**Occlusion estimation network.** The optical flow $f_{0 \to t}^L$ and $f_{1 \to t}^L$ are estimated using the Taylor network. This enables us to forward-warp the features of reference LR frames $I_0$ and $I_1$ to generate the spatial features at target time $t$. However, it is inevitable to encounter multiple source positions mapping to the same target pixels in the forward warping process. Taking inspiration from the work by Wang et al. Wang et al. (2018), we introduce an occlusion mask to alleviate this issue. Different objects generate event streams at varying frequencies due to differences in their speed, shape, and texture. When object A occludes object B, the resulting event stream predominantly reflects the characteristics of object A. This property of event streams allows us to effectively distinguish between foreground and background objects. To utilize this variability, we leverage event frequency information encapsulated in the event surface format. This format includes both the frequency and polarity information of the event streams, enabling us to estimate occlusion information for pixels that map to the same position. Consequently, we can assign weights to these pixels, enhancing the

visual quality of the warped images. We have developed an occlusion estimation network based on the U-Net architecture. This network takes event surfaces and image features as inputs and outputs occlusion features. Structure details of the occlusion estimation network is provided in the Appendix.

### 3.3 Loss Function

We employ an end-to-end supervised training approach for our proposed EvTaylor-Net, using the high-resolution images $I_t^H$ as the ground truth. The overall loss function is defined as:

$$\mathcal{L} = \mathcal{L}_{char}(\hat{I}_t^H, I_t^H), \tag{12}$$

where $\mathcal{L}_{char}$, known as the Charbonnier loss Lai et al. (2017), is calculated as: $\mathcal{L}_{char}(\hat{x}, x) = \sqrt{\|\hat{x} - x\|^2 + \epsilon^2}$. Here, the parameter $\epsilon$ is set to $10^{-3}$.

## 4 Experiment

### 4.1 Experiment Setup

**Implementation details.** Our S-INR and ST-INR modules follow the design of MoTIF Chen et al. (2023), using a 3-layer SIREN network Sitzmann et al. (2020) with hidden dimensions of 64, 64, and 256. The number of bins $B$ for event stream representation is set to 8. Similar with Chen et al. (2022b; 2023), we employ a two-stage training strategy for the model training. In the first stage of training, the complete model is trained with a spatial upscaling factor of $\times 4$ for 450,000 iterations, in conjunction with cosine annealing to dynamically adjust the learning rate from $10^{-4}$ to $10^{-7}$ every 150,000 iterations. In the second stage, we train the network for 150,000 iterations with randomly sampled scales from a uniform distribution $U(1, 4)$, starting with an initial learning rate $10^{-5}$. We utilize the Adam optimizer with hyperparameters $\beta_1 = 0.9$ and $\beta_2 = 0.999$ in the training process. The LR input size is randomly cropped to $32 \times 32$ during the training process. Data augmentation methods, including random rotations and horizontal flips, are applied in the training process. Our framework is trained with a batch size of 24 using two NVIDIA 3090 GPU cards. Additionally, the derivative order in the Taylor network is set to 3.

**Dataset.** We trained our model on the Adobe240 dataset Su et al. (2017). Initially, we performed bicubic downsampling by a factor of $\times 4$ on the original Adobe240 videos to generate the LR videos. Subsequently, we synthesized event streams based on the LR videos using the off-the-shelf v2e method Hu et al. (2021). During the training phase, the $1st$ and $9th$ LR frames, along with the concurrent event stream, serve as inputs to the model. During random spatial scale training, we use bicubic interpolation to downsample the 9 consecutive HR frames (i.e, $1st, 2nd, ..., 9th$) to the target size scale for model supervision. We use the Adobe240 dataset Su et al. (2017) and the GoPro Nah et al. (2017) dataset for the evaluation of STVSR tasks.

**Evaluation metrics.** In our evaluation, we compute metrics including Peak Signal-to-Noise Ratio (PSNR) and Structural Similarity Index (SSIM). Specifically, we calculate PSNR and SSIM on the Y channel of the YCbCr color space for the Adobe240 and GoPro datasets, consistent with previous works Chen et al. (2022b; 2023).

### 4.2 Quantitative Evaluation

**Comparison of STVSR.** On the Adobe240 Su et al. (2017) and GoPro Nah et al. (2017) datasets, we calculate the metrics for $\times 8$ temporal interpolation and single-frame interpolation at a spatial scale of $\times 4$. Consistent with previous works Chen et al. (2022b; 2023), we refer to these two metrics as -*Average* and -*Center* in Tab. 1, respectively. We evaluate different STVSR models, encompassing both one-stage and two-stage STVSR methods. For the two-stage STVSR methods, we employ the frame based methods including RIFE Huang et al. (2022), IFRNet Kong et al. (2022), and the event-based method, TimeLens Tulyakov et al. (2021) for VFI. The VSR methods we compared include bicubic interpolation, EDVR Wang et al.

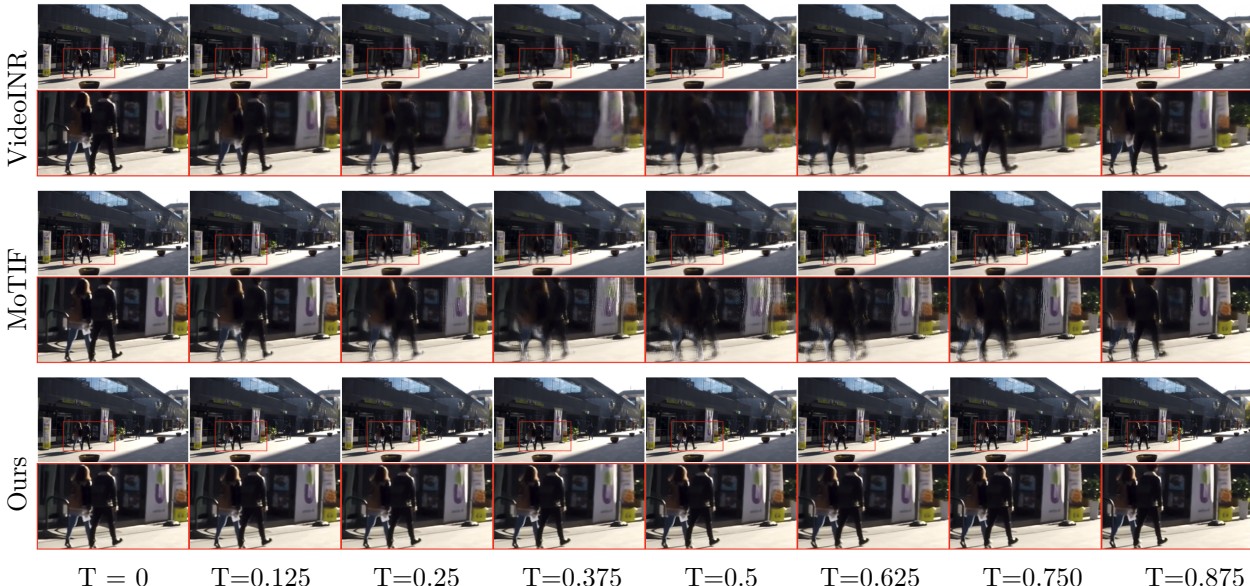

Figure 4: Visual comparison of C-STVSR methods on the GoPro dataset (time ×8, space ×4). Zoom in for better visualization.

(2019), BasicVSR++ Chan et al. (2022). For the one-stage STVSR, we compared our method with F-STVSR methods including Zooming SlowMo Xiang et al. (2020), TMNet Xu et al. (2021b), and also with C-STVSR methods consist of VideoINR Chen et al. (2022b) and MoTIF Chen et al. (2023). Some results are excerpted from Chen et al. (2022b) due to the same training dataset. The results in Tab. 1 indicate that our method exhibits superior performance on the both Adobe240 and GoPro datasets on the F-STVSR tasks. Fig. 4 further illustrates the visual effect of the cutting-edge C-STVSR methods on the GoPro dataset, with a spatial scale of ×8 and a temporal scale of ×4. It is observable that the frame interpolation quality achieved by EvTaylor-Net exceeds that of VideoINR Chen et al. (2022b) and MoTIF Chen et al. (2023).

To validate the continuous spatial and temporal super-resolution performance of our method, we conducted evaluation experiments with most spatiotemporal scaling factors not encountered during the training phase. Tab. 2 presents the quantitative results for these out-of-distribution scales on the GoPro dataset. We compare our method with both VFI+SR combinations—SuperSloMo Jiang et al. (2018)+LIIF Chen et al. (2021) and DAIN Bao et al. (2019a)+LIIF Chen et al. (2021)—and representative STVSR methods, including TMNet Xu et al. (2021b), VideoINR Chen et al. (2022b), and MoTIF Chen et al. (2023). It demonstrates that, although our method slightly underperforms compared to MoTIF Chen et al. (2023) in scenarios with time scale ×6 and space scale ×6, as well as time scale ×6 and space scale ×12, our method outperforms MoTIF Chen et al. (2023) in the vast majority of instances. As the temporal scaling factor increases, our results significantly surpass those of MoTIF Chen et al. (2023), clearly illustrating EvTaylor-Net's superior capability in handling temporal information. Fig. 5 presents visual results of one-stage C-STVSR methods on the GoPro dataset, where the HR frames produced by our method exhibit higher clarity and are devoid of motion blur. These findings underscore the strong generalization ability of our approach.

## 4.3 Ablation and Analysis Study

**Impact of the sub-networks.** To assess the contribution of the Taylor network, we conduct an experiment where the Taylor network was replaced with a U-Net architecture for the purpose of optical flow estimation. Furthermore, the efficacy of the occlusion estimation network is also put to test. For this, we remove the occlusion estimation network and adopt the average splatting method in the forward warping process. For a fair comparison, we train these two models and our complete model on the Adobe240 dataset with the random samples scales of a uniform distribution $U(1, 4)$ for 150,000 iterations. Tab. 3 showcases that with the same training strategy, the performance of our complete model is better than the model without occlusion

Table 1: Quantitative comparison on GoPro Nah et al. (2017) and Adobe240 Su et al. (2017) datasets. Red, blue indicate the best, the second best performance, respectively. Quality metrics: PSNR/SSIM.

| VFI Method | SR Method | GoPro-Center | GoPro-Average | Adobe-Center | Adobe-Average | Parameters (Million) |
|---|---|---|---|---|---|---|
| RIFE | Bicubic | 28.47/0.8296 | 28.10/0.8210 | 27.40/0.7863 | 27.09/0.7783 | 10.71 |
| RIFE | EDVR | 30.31/0.8745 | 29.59/0.8651 | 29.62/0.8592 | 29.09/0.8514 | 10.71+20.70 |
| RIFE | BasicVSR++ | 29.83/0.8669 | 29.20/0.8579 | 29.47/0.8530 | 28.97/0.8459 | 10.71+7.32 |
| IFRNet | Bicubic | 28.04/0.8168 | 27.42/0.8020 | 26.93/0.7697 | 26.39/0.7555 | 4.96 |
| IFRNet | EDVR | 29.68/0.8584 | 28.57/0.8398 | 28.74/0.8315 | 27.80/0.8123 | 4.96+20.70 |
| IFRNet | BasicVSR++ | 28.32/0.8338 | 27.60/0.8189 | 27.75/0.8106 | 27.07/0.7954 | 4.96+7.32 |
| TimeLens | Bicubic | 27.55/0.8044 | 26.23/0.7746 | 27.18/0.7765 | 26.61 /0.7603 | 79.2 |
| TimeLens | EDVR | 28.86/0.8361 | 26.63/0.7933 | 28.95/0.8376 | 27.79/0.8132 | 79.2+20.70 |
| TimeLens | BasicVSR++ | 28.42/0.8254 | 26.36/0.7837 | 28.68/0.8287 | 27.60/0.8049 | 79.2+7.32 |
| Zooming SlowMo | | 30.69/0.8847 | -/- | 30.26/0.8821 | -/- | 11.10 |
| TMNet | | 30.14/0.8692 | 28.83/0.8514 | 29.41/0.8524 | 28.30/0.8354 | 12.26 |
| VideoINR-fixed | | 30.73/0.8850 | -/- | 30.21/0.8805 | -/- | 11.31 |
| VideoINR | | 30.26/0.8792 | 29.41/0.8669 | 29.92/0.8746 | 29.27/0.8651 | 11.31 |
| MoTIF | | 31.04/0.8877 | 30.04/0.8773 | 30.63/0.8839 | 29.82/0.8750 | 12.55 |
| Ours | | 32.12/0.9148 | 31.84/0.9122 | 30.75/0.8894 | 30.43/0.8865 | 34.19 |

Table 2: Quantitative comparison for out-of-distribution scales on the GoPro dataset. Red, blue indicate the best, the second best performance, respectively. Quality metrics: PSNR/SSIM.

| Time | Space | SuperSloMo +LIIF | DAIN + LIIF | TMNet | VideoINR | MoTIF | Ours |
|---|---|---|---|---|---|---|---|
| ×6 | ×4 | 26.70/0.7988 | 26.71/0.7998 | 30.49/0.8861 | 30.78/0.8954 | 31.56/0.9064 | 32.31/0.9190 |
| ×6 | ×6 | 23.47/0.6931 | 23.36/0.6902 | - | 25.56/0.7671 | 29.36/0.8505 | 29.10/0.8468 |
| ×6 | ×12 | 21.92/0.6495 | 22.01/0.6499 | - | 24.02/0.6900 | 25.81/0.7330 | 25.23/0.7189 |
| ×12 | ×4 | 25.07/0.7491 | 25.14/0.7497 | 26.38/0.7931 | 27.32/0.8141 | 27.77/0.8230 | 30.72/0.8937 |
| ×12 | ×6 | 22.91/0.6783 | 22.92/0.6785 | - | 24.68/0.7358 | 26.78/0.7908 | 28.40/0.8315 |
| ×12 | ×12 | 21.61/0.6457 | 21.78/0.6473 | - | 23.70/0.6830 | 24.72/0.7108 | 25.02/0.7145 |
| ×16 | ×4 | 24.42/0.7296 | 24.20/0.7244 | 24.72/0.7526 | 25.81/0.7739 | 25.98/0.7758 | 29.41/0.8668 |
| ×16 | ×6 | 23.28/0.6883 | 22.80/0.6722 | - | 23.86/0.7123 | 25.34/0.7527 | 27.69/0.8149 |
| ×16 | ×12 | 21.80/0.6481 | 22.22/0.6420 | - | 22.88/0.6659 | 23.88/0.6923 | 24.79/0.7097 |
| Parameters (Million) | | 39.61+22.32 | 24.0+22.32 | 12.26 | 11.31 | 12.55 | 34.19 |

estimation network or without the Taylor network, which validates the effectiveness of the Taylor network and the occlusion estimation network.

**Absolute time vs. Relative time.** EvTaylor-Net maps the time between $I_0^L$ and $I_1^L$, i.e, absolute time rather than relative time, into $B$ time bins to represent the event voxel grid. We train the proposed framework with relative time manner on the Adobe240 training dataset for 150,000 iteration with a random sample scale from $U(1,4)$. The comparative results are presented in Tab. 3, which demonstrate that our complete model, utilizing absolute time, outperforms the one relying on relative time.

**The effectiveness of residual features.** In the decoder, we take the residual features $F_{(0,1)}^H$ and the target feature $F_t^H$, as well as time $t$ as input. To evaluate the impact of the residual features in the decoding process, we train the EvTaylor-Net without the residual features on the Adobe240 dataset for 150,000 iterations with a random sampled scales from a uniform distribution $U(1,4)$. The comparative results are detailed in Tab. 3. These findings underscore the significance of incorporating residual features in the decoder.

**Analysis on the derivative order in the Taylor network.** The proposed EvTaylor-Net enables the optical flow estimation considering the $nth$ order motion. When the derivative order $n$ specified from 1 to 3, our model can be degraded as the linear model, quadratic model, and cubic model respectively, i.e, our method not only covers all previous motion assumptions, but also is more flexible, i.e, without modifying any network structure. In the aforementioned experiment, we maintain a derivative order of n = 3. To further evaluate the influence of varying derivative orders in the Taylor network, we conduct comparative experiments by specifying the derivative order (n = 1, 2, 3) in the Taylor network. The comparison on the F-STVSR tasks

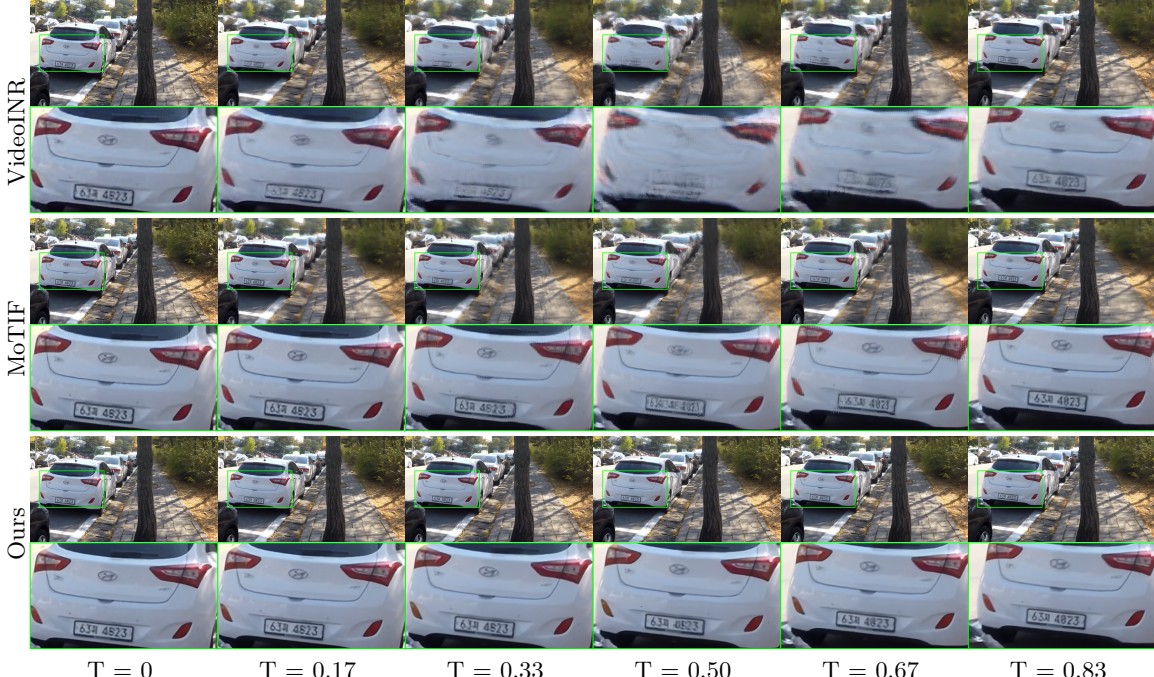

Figure 5: Visual results of different C-STVSR methods on the GoPro dataset, with interpolation times being out-of-distribution (time ×6, space ×4).

Table 3: Quantitative results of ablation studies. Quality metrics: PSNR / SSIM.

| Our Methods | GoPro-Center | GoPro-Average | Adobe-Center | Adobe-Average |
|---|---|---|---|---|
| w/o Occl. | 30.12 / 0.8738 | 29.70 / 0.8671 | 29.18 / 0.8510 | 28.81 / 0.8443 |
| w/o Tayl. | 30.66 / 0.8844 | 30.26 / 0.8784 | 29.61 / 0.8576 | 29.21 / 0.8525 |
| w/ relative time | 30.66 / 0.8839 | 30.21 / 0.8788 | 29.59 / 0.8576 | 29.12 / 0.8519 |
| w/o res. | 30.78 / 0.8857 | 30.35 / 0.8805 | 29.68 / 0.8581 | 29.26 / 0.8525 |
| complete model | **31.33 / 0.8989** | **31.04 / 0.8962** | **29.90 / 0.8655** | **29.53 / 0.8620** |

of the frameworks under different derivative orders are shown in Tab. 4, indicating that the derivative order n = 3 presents the best evaluation performance. The performance comparison of EvTaylor-Net with different derivative orders on the C-STVSR tasks is presented in Tab. 5. It is evident that our model achieves optimal performance in all spatiotemporal scales when n = 3, as indicated by the highest PSNR and SSIM values.

Table 4: Quantitative evaluation of EvTaylor-Net with different derivative orders on the F-STVSR tasks. Quality metrics: PSNR / SSIM.

| Our Methods | GoPro-Center | GoPro-Average | Adobe-Center | Adobe-Average |
|---|---|---|---|---|
| n=1 | 32.01 / 0.9126 | 31.75 / 0.9094 | 30.57 / 0.8856 | 30.23 / 0.8821 |
| n=2 | 32.06 / 0.9129 | 31.81 / 0.9105 | 30.68 / 0.8871 | 30.41 / 0.8846 |
| n=3 | **32.12 / 0.9148** | **31.84 / 0.9122** | **30.75 / 0.8894** | **30.43 / 0.8865** |

## 5  Conclusion

In this paper, we introduce EvTaylor-Net to simultaneously address the challenges of continuous space-time video super-resolution. Our approach harnesses the full potential of event streams to estimate optical flow with high precision. By utilizing the Taylor expansion approximation, the EvTaylor-Net accounts for higher-order motion dynamics, estimating more accurate optical flow. During the forward warping process, we confront the common issue where multiple source pixels converge on a single target position. To address this, we

Table 5: Quantitative evaluation of EvTaylor-Net with different derivative orders on the C-STVSR tasks. Quality metrics: PSNR / SSIM. **Bold** indicates the best performance.

| Time | Space | n = 1 | n = 2 | n = 3 |
|------|-------|-------|-------|-------|
| $\times$ 6 | $\times$ 4 | 32.19/0.9160 | 32.22/0.9165 | **32.31/0.9190** |
| $\times$ 6 | $\times$ 6 | 28.92/0.8418 | 29.98/0.8441 | **29.10/0.8468** |
| $\times$ 6 | $\times$ 12 | 25.10/0.7177 | 25.13/0.7184 | **25.23/0.7189** |
| $\times$ 12 | $\times$ 4 | 30.61/0.8907 | 30.65/0.8914 | **30.72/0.8937** |
| $\times$ 12 | $\times$ 6 | 28.13/0.8286 | 28.16/0.8301 | **28.40/0.8315** |
| $\times$ 12 | $\times$ 12 | 24.90/0.7125 | 25.00/0.7134 | **25.02/0.7145** |
| $\times$ 16 | $\times$ 4 | 29.32/0.8637 | 29.35/08644 | **29.41/0.8668** |
| $\times$ 16 | $\times$ 6 | 27.41/0.8068 | 27.47/0.8077 | **27.69/0.8149** |
| $\times$ 16 | $\times$ 12 | 24.54/0.7052 | 24.60/0.7062 | **24.79/0.7097** |

introduce an occlusion estimation network that leverages event data. This network is designed to predict masks that serve as key metrics in the softmax splatting operation, enhancing the quality of image warping. The comprehensive experiments on the public datasets we conducted clearly demonstrate the superiority of our newly proposed EvTaylor-Net in handling these complex tasks, underscoring the potential of event streams in the field of continuous space-time video super-resolution.

**Limitation.** Like other continuous space-time super-resolution models, our model also requires a prolonged period of training. How to enhance training efficiency so that the model still retains excellent performance under continuous spatio-temporal scaling remains an open question in the continuous space-time super-resolution domain.

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

# A    Appendix

## A.1    Additional Network Details

**Feature extractors for voxel grids and event surfaces.** To extract the features from the event voxel grids and event surface, we adopt the feature extractors as illustrated in Fig. 6. The feature extractor, depicted in Fig. 6 (a), processes the forward event voxel grids $Ev_{0\to1}$, which consist of 16 channels. This extractor outputs features with an expanded 64 channels. It's noteworthy that the same extraction module is also employed for the backward event voxel grids. Further, as Fig. 6 (b) shows, the feature extractor for event surfaces handles the forward event surfaces $Es_{0\to1}$, which have 32 channels, converting them into features with 64 channels. This module is similarly used for processing backward event surfaces. These extracted features from both the event voxel grids and event surfaces are then inputted into two networks: the Taylor network and the occlusion estimation network, for subsequent processing.

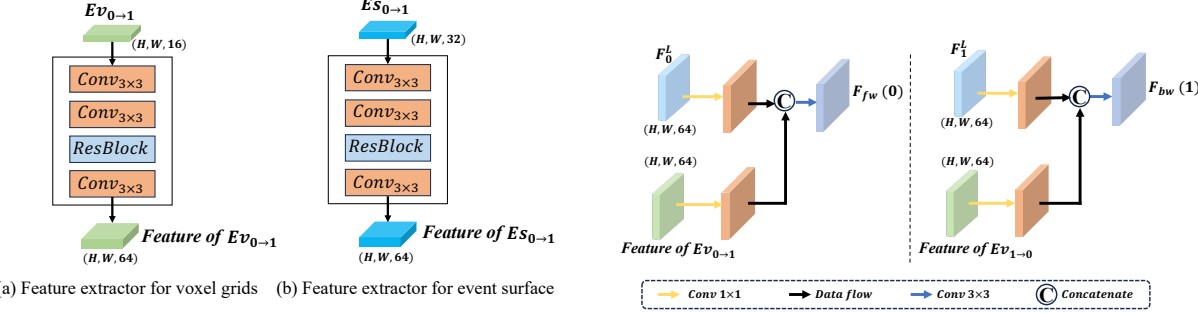

Figure 6: Feature extraction modules for event voxel grids and event surfaces.

Figure 7: Network architecture of the fusion network.

**Structure of the fusion network.** In the Taylor network,we integrate the features of event voxel grids with image features using a fusion network. For estimating the forward optical flow $f_{0\to t}^L$, the fusion network's inputs are $F_0^L$ and features of $Ev_{0\to1}$. As for the backward optical flow $f_{1\to t}^L$, the inputs comprise $F_1^L$ and features of $Ev_{1\to0}$. The detailed architecture of this fusion network is depicted in Fig. 7. Initially, the network applies a convolutional layer with a 1×1 kernel for independent feature extraction from event voxel grids and image features. This is followed by a 3×3 kernel convolutional layer, designed to effectively merge these extracted features. Finally, the fusion network outputs the motion features, including the $F_{fw}(0)$ and $F_{bw}(1)$, at the timestamp of reference frames, which servers as the input of the subsequent derivative network G.

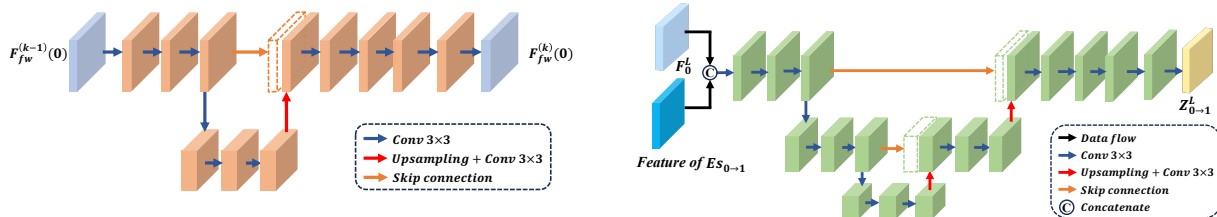

Figure 8: Network architecture of the derivative network G.

Figure 9: Network architecture of the occlusion estimation network.

**Structure of the derivative network G.** Within Taylor network, we implement the derivative network $G$ using a U-Net architecture. This network plays a critical role in estimating the accurate optical flow. Specifically, as illustrated in Fig. 8, for the forward optical flow $f_{0\to t}^L$ estimation, the derivative network $G$ takes $F_{fw}^{(k-1)}(0)$ as its input and computes its derivative, resulting in $F_{fw}^{(k)}(0)$. Similarly, for the estimation of the backward optical flow $f_{1\to t}^L$, the derivative network $G$ processes the features $F_{bw}^{(k-1)}(1)$ and outputs the derivative $F_{bw}^{(k)}(1)$.

**Structure of the occlusion estimation network.** To address the challenge where multiple source pixels map to the same target position, we have developed an occlusion estimation network. This network is designed to compute masks that assign weights to these source pixels, thereby enhancing the visual quality of the warped images. Fig. 9 illustrates the network structure of the occlusion estimation network, estimating the forward mask $Z_{0 \to 1}^L$. This process involves using the features of event surface $Es_{0 \to 1}$ and the image features $F_0^L$ as input. The network then estimates the mask features $Z_{0 \to 1}^L$. Regarding the estimation of $Z_{1 \to 0}^L$, the occlusion estimation network similarly processes the features of $Es_{1 \to 0}$ and $F_1^L$, and outputs the mask $Z_{1 \to 0}^L$.

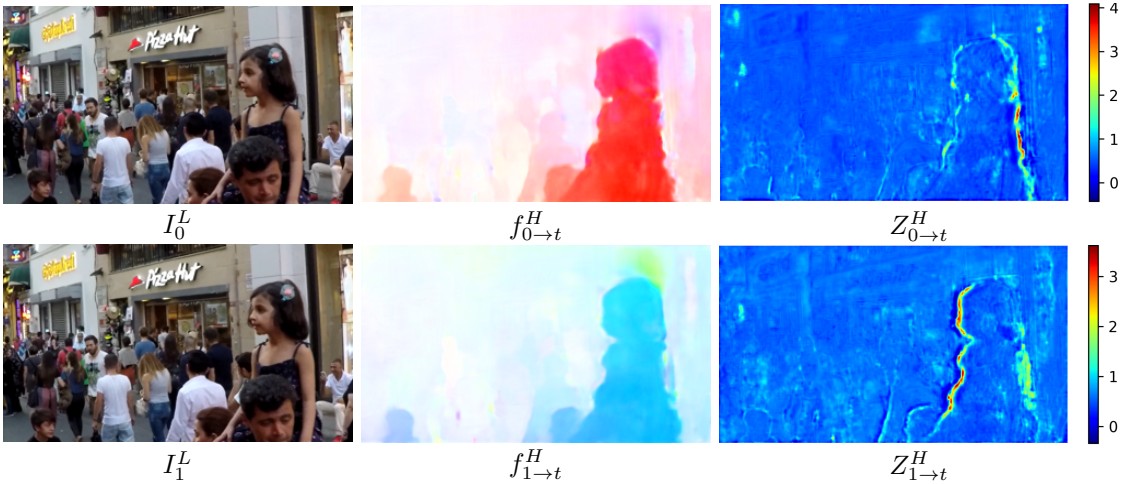

$I_0^L$ $\qquad\qquad\qquad\qquad$ $f_{0 \to t}^H$ $\qquad\qquad\qquad\qquad$ $Z_{0 \to t}^H$

$I_1^L$ $\qquad\qquad\qquad\qquad$ $f_{1 \to t}^H$ $\qquad\qquad\qquad\qquad$ $Z_{1 \to t}^H$

Figure 10: Visualization of flows and masks (space $\times 4$, $t = 0.5$).

**Intermediate visualization results:** As shown in the Fig. 10, we visualize the final optical flow maps $f_{0 \to t}^H$ and $f_{1 \to t}^H$, and the estimated occlusion masks $Z_{0 \to t}^H$ and $Z_{1 \to t}^H$. The primary motion in images $I_0^L$ and $I_1^L$ is attributed to the girl moving from left to right. It is evident that the optical flow estimated by our network accurately represents this motion. The main occlusion occurs in the area surrounding the girl, which is accurately captured by EvTaylor-Net.

**Additional visual results.** For a more nuanced comparison of spatial super-resolution impacts, Fig. 11 presents a visual evaluation of the state-of-the-art C-STVSR methods on the Adobe240 dataset, the recovered HR frames by our method have a better visual quality. Fig.12 shows visual comparisons on the Adobe240 dataset, while Fig.13 shows visual quality results on the GoPro dataset. The spatial scale of the above results is $\times 4$ and the time scale is $\times 8$. In addition, Fig.14 shows the visual results when the spatial scale is $\times 4$ and the time scale is $\times 6$. These visual comparisons all demonstrate the superiority of our approach.

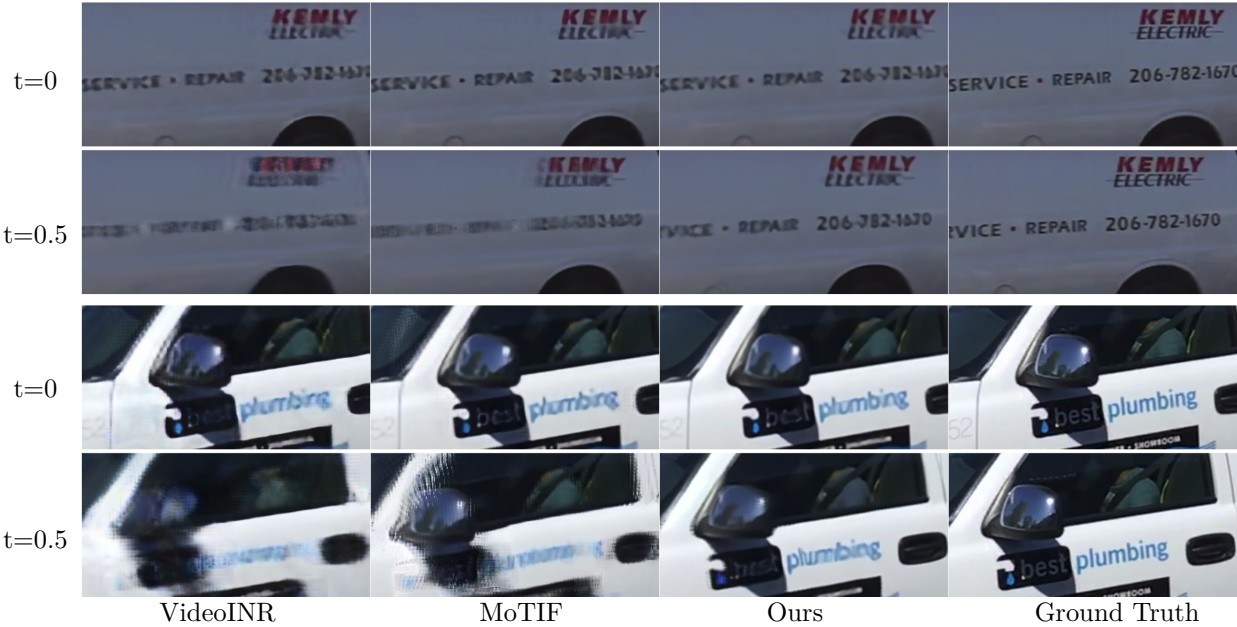

Figure 11: Visual effects of C-STVSR methods on the Adobe240 dataset (time ×8, space ×4). We crop 160×320 patches for visualization.

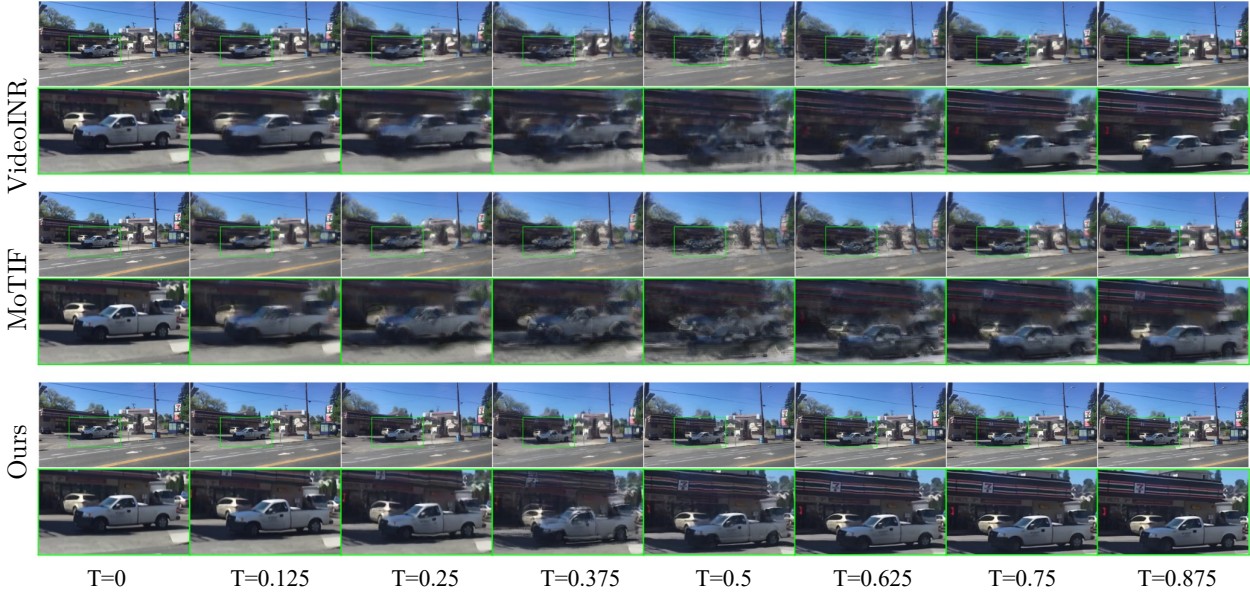

Figure 12: Visual effects of C-STVSR methods on the Adobe240 dataset (time ×8, space ×4). Zoom in for better visualization.

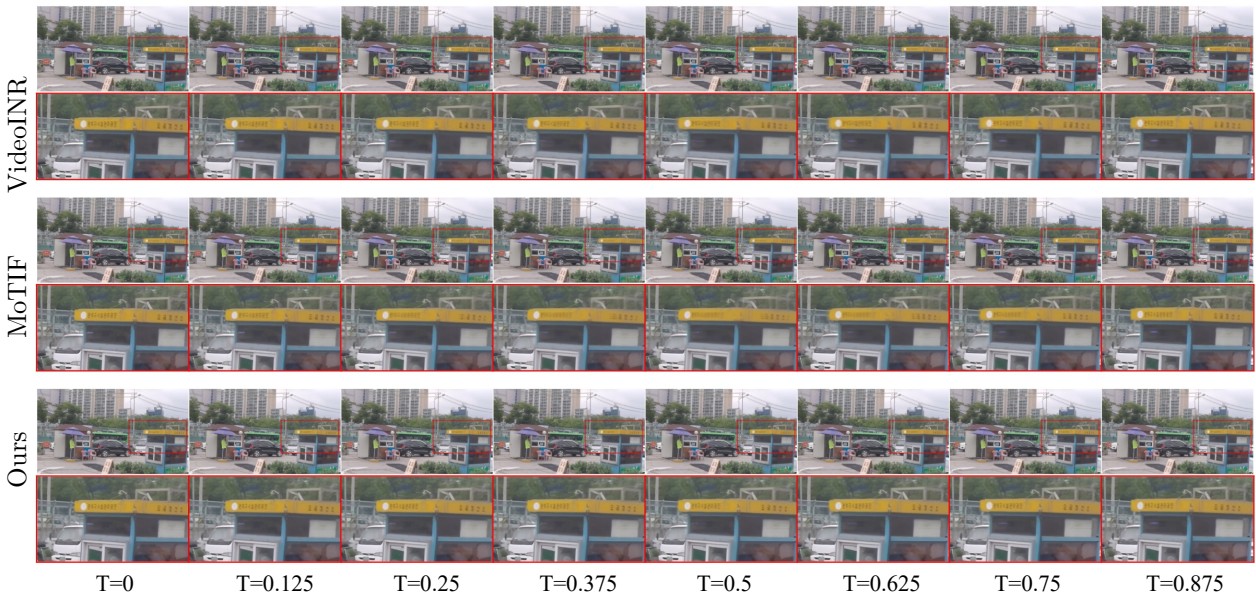

Figure 13: Visual effects of C-STVSR methods on the GoPro dataset (time ×8, space ×4). Zoom in for better visualization.

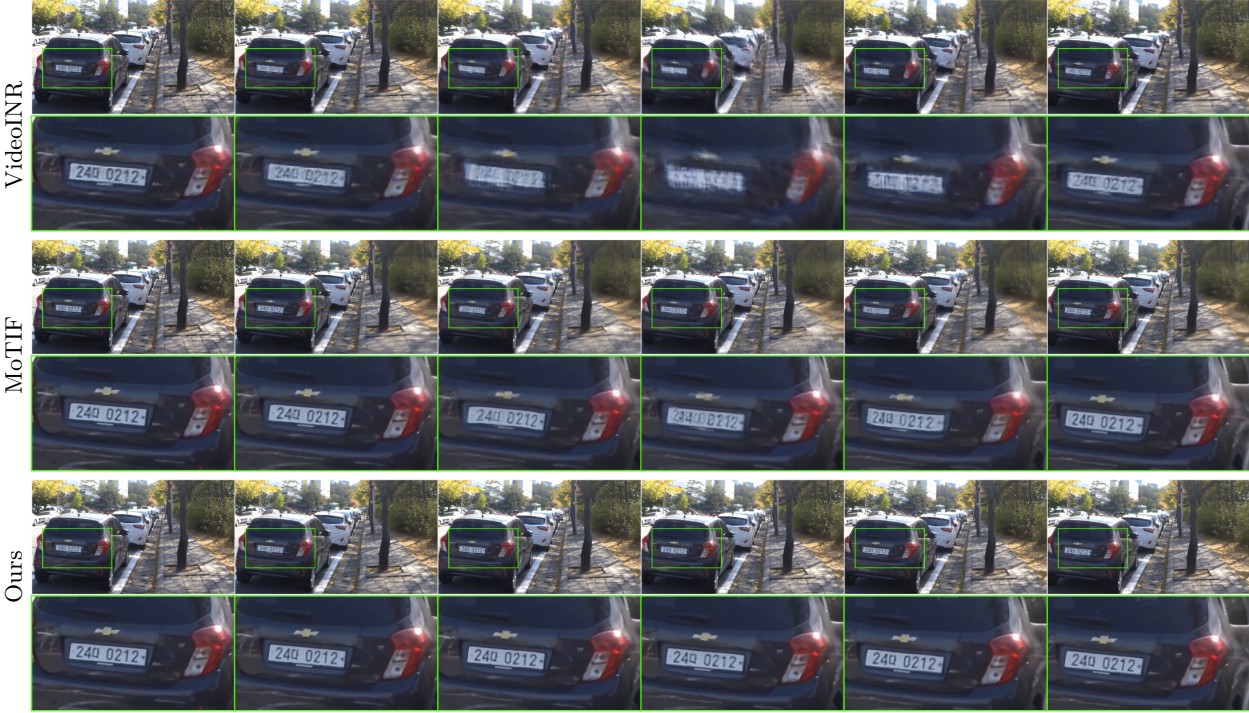

Figure 14: Visual results of different C-STVSR methods on arbitrary frame interpolation, with interpolation times being out-of-distribution on the GoPro dataset (time ×6, space ×4).

