# OpenReview forum: "Learning High-Order Motion Patterns from Event Stream for Continuous Space-Time Video Super-Resolution"
_TMLR — Rejected by TMLR_

### Review · Reviewer_YX3s · 2025-11-29

**Summary Of Contributions:**

The paper proposes EvTaylor-Net, a novel framework for Continuous Space-Time Video Super-Resolution (C-STVSR) that integrates neuromorphic event streams with conventional video frames to address the limitations of discrete imaging systems.

Key Contributions:

1. It is the first work to utilize event streams for the C-STVSR task. By partitioning events into "Voxel Grids," the method bridges the temporal "blind spots" between shutter exposures, allowing the model to measure rather than hallucinate inter-frame motion.

2. The authors introduce a theoretical innovation by modeling motion via an $n$-th order Taylor expansion. A specialized network approximates high-order derivatives (velocity, acceleration, jerk) to construct a forward-only optical flow. This overcomes the limitations of linear assumptions and the ill-posed nature of backward warping found in prior INR-based methods.

3. The method leverages event firing frequencies (aggregated into "Event Surfaces") as a proxy for depth and importance, generating occlusion masks that significantly reduce ghosting artifacts during warping.

Strengths:

1. Outperforms existing methods (VideoINR, MOTIF) significantly, particularly in extreme temporal upsampling scenarios (up to $\times 16$).

2. The rejection of backward warping in favor of a causal, kinematic-based forward model is physically sound and novel.

Weaknesses:

1. The model has roughly $3\times$ the parameter count of baselines, with no efficiency metrics (FLOPs/FPS) provided.

2. Validation is performed strictly on simulated (v2e) data, leaving robustness to real-world sensor noise unproven.

**Audience:**

Yes

**Audience Explanation:**

Several TMLR sub-communities would find the paper compelling:

1. Computer Vision & Computational Photography: Researchers working on video restoration, super-resolution, and Implicit Neural Representations (INR) will find the method for stabilizing continuous-time rendering highly valuable.

2. Neuromorphic Engineering: The paper demonstrates a "killer app" for event cameras, showing they can achieve reconstruction quality impossible with frames alone. This provides a strong argument for the adoption of DVS sensors in standard imaging pipelines.

3. Deep Learning Theory (Inductive Biases): The architecture serves as a case study for embedding mathematical priors (Taylor series) into deep networks. This is relevant to readers interested in Physics-Informed Neural Networks (PINNs) and structured latent spaces.

4. Robotics: The ability to recover high-speed motion details (acceleration/jerk) is relevant for navigation and tracking in autonomous systems.

**Broader Impact Concerns:**

1. The core capability of this work is to recover high-fidelity visual details (text, faces, identifiers) from low-resolution, motion-blurred inputs using event data. This negates the privacy preservation incidental to low-quality surveillance footage. Individuals who might assume they are unidentifiable due to distance or speed could be re-identified using this method.

2. The focus on recovering "High-Order Motion" (acceleration, jerk) makes this technology particularly effective for tracking hyper-agile objects. This has clear dual-use potential in military applications for targeting and tracking projectiles or drones that defy standard linear motion models.

3. The authors should explicitly acknowledge these capabilities in a Broader Impact Statement, noting the potential for advanced forensic reconstruction and high-speed tracking beyond simple video enhancement.

**Claims And Evidence:**

Yes

**Claims Explanation:**

The claims are generally supported, with a few suggestions:

1. The claim of outperforming SOTA is supported by rigorous benchmarking on the GoPro and Adobe240 datasets. Specifically, the method achieves a 1.08 dB gain over the nearest competitor (MOTIF) on the GoPro-Center benchmark, which is a statistically significant margin in image restoration.

2. The claim that the Taylor expansion better captures continuous dynamics is validated by the performance gap widening at high interpolation rates. At $\times 16$ temporal upsampling, EvTaylor-Net exceeds MOTIF by nearly 3.5 dB.  This provides compelling evidence that high-order terms (acceleration/jerk) are necessary for long-range temporal coherence.

3. The ablation studies explicitly validate the theoretical design, showing that increasing the Taylor order from $n=1$ (linear) to $n=3$ monotonically improves PSNR.

Suggestions:

The evidence for "efficiency" is missing (no FPS/FLOPs data), and the claim of robustness to "Event Streams" is inferred from synthetic simulators (v2e) rather than proven on noisy, real-world DVS hardware.

**Requested Changes:**

1. The text must be revised to clarify that the network learns approximations of high-order motion features that function as Taylor coefficients, rather than mathematically "producing derivatives." The current phrasing implies a symbolic exactness that neural networks do not inherently possess without specific physical constraints.

2. The authors must provide Inference Time (FPS) and FLOPs comparisons against baselines (VideoINR, MOTIF). Given the significant increase in parameters ($34M$ vs $11M$), proving practical feasibility is essential. Please clarify if the Taylor network runs once per frame pair or must be re-evaluated for every target timestamp $t$.

3. Include a brief discussion or failure case analysis regarding the assumption that event frequency equals occlusion depth. The authors should acknowledge potential failures in scenarios with high-contrast backgrounds versus textureless foregrounds.

4. Add a discussion on the transferability of the model from v2e synthetic data to real-world event sensors. Specifically, address how real-world sensor noise (hot pixels, refractory noise) might affect the derivative estimation.

5. Explicitly contrast the Taylor expansion approach with spline-based methods used in other event-based VFI literature (e.g., TimeLens++) to highlight the theoretical advantages of the polynomial approach.

---

### Review · Reviewer_sChG · 2025-12-18

**Summary Of Contributions:**

The paper proposes a framework for continuous space-time video super-resolution that leverages event streams to handle complex motion. Key contributions:
- Unlike existing methods that assume linear or quadratic motion, the authors use a Taylor expansion approximation to model $n^{th}$-order motion (specifically 3rd order in experiments) from event data.
- The paper introduces a Taylor Network that fuses image and event features to estimate bi-directional forward optical flow.
- An occlusion network is designed to generate importance masks based on event frequencies, addressing the many-to-one mapping problem in forward warping.
- The model uses S-INR and ST-INR to achieve arbitrary upscaling in both space and time.

**Strengths:**
- The shift from simple linear motion assumptions to a higher-order Taylor expansion seems to make sense.
- The paper provides a clear justification for using event streams to solve the inaccuracies of frame-only optical flow estimators.
- Quantitative results on benchmark datasets like Adobe240 and GoPro demonstrate that EvTaylor-Net generally outperforms recent methods.
- The integration of INR allows the model to handle out-of-distribution scales not seen during training.

**Weaknesses:**
- The experiments primarily use the v2e method to synthesize event streams from the Adobe240 dataset. It is unclear how the model performs on real-world, noisy event data from actual event cameras.
- While the authors use a 3rd-order derivative, there is no detailed ablation study exploring how different values of $n$ affect the trade-off between accuracy and performance.

**Additional Comments:**

I am not an expert in this specific field. My review focuses more on the clarity of the methodology and the consistency of the experimental evidence presented.

**Audience:**

Yes

**Audience Explanation:**

The integration of event-based vision with implicit neural representations should be of interest to some individuals in TMLR's audience.

**Broader Impact Concerns:**

I have no concerns.

**Claims And Evidence:**

Yes

**Claims Explanation:**

I believe the experiments conducted on the two datasets, together with the out-of-distribution scaling tests and ablation studies, provide supporting evidence for the claims made in the paper. However, it would be important for the authors to address the concerns raised below.

**Requested Changes:**

- Provide more intuition or visual evidence on how the 3rd-order term specifically helps in high-order motion scenarios compared to the 1st order.
- Include a small qualitative evaluation on a dataset captured with real event cameras (e.g., HQF or CED) to prove robustness against sensor noise.
- Add a table comparing FLOPs or runtime against MOTIF and VideoINR.
- Please proofread for grammatical and spelling errors. For example, in the first paragraph of Introduction, "perform" should be used instead of "preform."

---

### Review · Reviewer_t735 · 2025-12-31

**Summary Of Contributions:**

This paper introduces EvTaylor-Net, a framework for continuous space–time video super-resolution (C-STVSR) that enables arbitrary spatial and temporal upsampling of low-quality videos. It leverages event streams to provide high-temporal-resolution motion cues, overcoming limitations of frame-based optical flow. A Taylor Network models motion via high-order Taylor expansion to estimate forward flow at arbitrary timestamps, while an event-driven occlusion module resolves pixel collisions. Implicit Neural Representations (INRs) ensure continuous space–time reconstruction.

**Additional Comments:**

**Reproducibility & Missing Details**

To ensure the community can build upon this work, I request that the authors address the following:

1) The training relies on synthetic events generated using the v2e toolbox. It would be helpful if the authors could specify the exact parameter settings, particularly the positive and negative contrast thresholds ($\theta$) and any injected temporal noise, as these choices directly affect the density and quality of the event representations.

2) While the model size (34.19M parameters) is reported, the use of Implicit Neural Representations and higher-order derivative computations may introduce nontrivial computational overhead. Providing average inference time (e.g., milliseconds or FPS) on a standard GPU such as an RTX 3090 for a $\times 4$ spatial and $\times 8$ temporal upscaling setting would help clarify the practical efficiency of the method.

3) The paper mentions that training requires a “prolonged period,” but does not quantify this cost. Reporting the total training time in GPU hours or days would allow readers to better assess the feasibility of reproducing or extending the approach.

4)  In Section 3.2, the authors introduce a new time partitioning scheme over the interval $[0,1]$. To ensure reproducibility, please provide explicit pseudocode or a code snippet describing how the normalized timestamps $t_i$ in Equation (6) are computed for voxel grid construction.

5) To strengthen the claimed contribution to the community, I strongly encourage the authors to release the source code for the ST-INR and S-INR modules, along with pre-trained models on Adobe240 and GoPro, which would greatly facilitate adoption and follow-up research.

**Audience:**

Yes

**Audience Explanation:**

Yes, the audience would likely find the findings of this paper highly relevant. This paper presents a rigorous approach to the challenging computer vision task of Continuous Space-Time Video Super-Resolution (C-STVSR) by integrating event-based sensing with implicit neural representations.

**Broader Impact Concerns:**

Please discuss the broader implications and potential limitations of the proposed approach in the paper.

**Claims And Evidence:**

Yes

**Claims Explanation:**

**Strength**

1) First work to integrate event streams into the C-STVSR setting. Leveraging the high temporal resolution of event cameras effectively addresses the limitations of frame-only motion estimation.

2) The proposed Taylor-based formulation enables motion modeling up to arbitrary order ($n \ge 1$), providing greater flexibility than prior approaches that rely on fixed linear, quadratic, or cubic motion assumptions.

3) The occlusion estimation module is well grounded in physical intuition, using event frequency statistics, which naturally depend on object speed, shape, and texture, to separate foreground and background, leading to improved forward warping.

4) EvTaylor-Net consistently outperforms strong baselines (e.g., VideoINR, MOTIF) on standard benchmarks such as Adobe240 and GoPro.

**Weakness**

1) EvTaylor-Net is nearly triple the size of MOTIF (12.55M), which may hinder deployment on edge devices.

2) The authors admit the model requires a "prolonged period of training," which limits experimental iteration.

3) The training and evaluation rely on events synthesized via v2e rather than real-world event sensor data, potentially introducing a domain gap.

**Requested Changes:**

**Questions**

1) The derivative network G is applied recursively to estimate higher-order motion terms. Since such recursion can lead to gradient instability, did the authors use any stabilization techniques (e.g., layer normalization or residual scaling) to ensure numerical stability for n=3?

2) While the ablation shows n=3 outperforming lower orders, higher-order Taylor expansions should, in principle, further improve accuracy. Why were n≥4 not explored, and is there a practical order limit imposed by event noise or voxel grid resolution?

3) The use of absolute time bins is argued to reduce ambiguity. How does the method handle highly non-uniform event distributions, especially when events are sparse in later time bins?

4) Since the occlusion mask is based on event frequency, how does the model handle fast-moving but textureless objects against textured backgrounds?

5) Are there any explicit constraints or auxiliary losses that encourage the first- and second-order derivatives to correspond to physical velocity and acceleration, or are these treated as unconstrained latent features?

6) Evaluate the pre-trained EvTaylor-Net on the DSEC or MVSEC datasets (which contain real event camera data) without fine-tuning. Quantitative results (PSNR/SSIM) on these datasets would confirm if the "Synthetic Domain Bias" noted in the weaknesses is a significant barrier to deployment.

7) Provide a table showing the Inference Time (ms) and GFLOPs as $n$ increases. This will help reviewers understand the trade-off between the marginal PSNR gains of $n=3$ and the increased latency of running the shared-weight U-Net $G$ multiple times.

---

### Decision · Action_Editor_wgmR · 2026-02-23

**Recommendation:** Reject

**Additional Comments:**

However, the authors didn't respond to the reviewers' comments.

**Audience:**

Yes

**Audience Explanation:**

The topic of continuous space-time video super-resolution is an important topic in the field.

**Claims And Evidence:**

No

**Claims Explanation:**

This paper proposes EvTaylor-Net, which performs a Taylor expansion approximation of the object motion function. However, the authors didn't respond to the reviewers' comments, which include extensive concerns about details and experiments.

**Resubmission Of Major Revision:**

The authors may consider submitting a major revision at a later time.